# Predictive Value of Soluble PD-1, PD-L1, VEGFA, CD40 Ligand and CD44 for Nivolumab Therapy in Advanced Non-Small Cell Lung Cancer: A Case-Control Study

**DOI:** 10.3390/cancers12020473

**Published:** 2020-02-18

**Authors:** Manuela Tiako Meyo, Anne Jouinot, Etienne Giroux-Leprieur, Elizabeth Fabre, Marie Wislez, Marco Alifano, Karen Leroy, Pascaline Boudou-Rouquette, Camille Tlemsani, Nihel Khoudour, Jennifer Arrondeau, Audrey Thomas-Schoemann, Hélène Blons, Audrey Mansuet-Lupo, Diane Damotte, Michel Vidal, François Goldwasser, Jérôme Alexandre, Benoit Blanchet

**Affiliations:** 1Drug Biology–Toxicology, Cochin Hospital, AP-HP, CARPEM, 75014 Paris, France; nihel.khoudour@aphp.fr (N.K.); michel.vidal@aphp.fr (M.V.); benoit.blanchet@aphp.fr (B.B.); 2UMR8038 CNRS, U1268 INSERM, Faculty of Pharmacy, Paris Descartes University, PRES Sorbonne Paris Cité, 75006 Paris, France; audrey.thomas@aphp.fr; 3Department of Medical Oncology, Cochin Hospital, AP-HP, Paris Descartes University, CARPEM, 75014 Paris, France; anne.jouinot@aphp.fr (A.J.); pascaline.boudou@aphp.fr (P.B.-R.); camille.tlemsani@aphp.fr (C.T.); jennifer.arrondeau@aphp.fr (J.A.); francois.goldwasser@aphp.fr (F.G.); jerome.alexandre@aphp.fr (J.A.); 4Institut Cochin, INSERM U1016, 75014 Paris, France; 5Department of Respiratory Diseases and Thoracic Oncology, APHP-AmbroiseParé Hospital and EA 4340 University Versailles-Saint Quentin en Yvelines, 92100 Boulogne, France; etienne.giroux-leprieur@aphp.fr; 6Department of Thoracic Oncology, Georges Pompidou European Hospital, AP-HP, 75015 Paris, France; elizabeth.fabre@aphp.fr; 7Department of Pneumology, Cochin Hospital, APHP, 75014 Paris, France; marie.wislez@aphp.fr; 8Department of Thoracic Surgery, Cochin Hospital, APHP, 75014 Paris, France; marco.alifano@aphp.fr; 9Department of Cyto-pathology, Cochin Hospital, AP-HP, 75014 Paris, France; karen.leroy@aphp.fr (K.L.); audrey.lupo@aphp.fr (A.M.-L.); diane.damotte@aphp.fr (D.D.); 10Department of Cyto-pathology, Georges Pompidou European Hospital, AP-HP, 75015 Paris, France; helene.blons@aphp.fr; 11Institut Cordeliers, INSERM U1147, 75006 Paris, France

**Keywords:** Immune checkpoint inhibitors, nivolumab, NSCLC, biomarkers, PD-1

## Abstract

A large interindividual variability has been observed in anti Programmed cell Death 1 (anti-PD1) therapies efficacy. The aim of this study is to assess the correlation of soluble PD-1 (sPD-1), soluble Programmed cell Death Ligand 1 (sPD-L1), Vascular Endothelial Growth Factor A (VEGFA), soluble CD40 ligand (sCD40L) and soluble CD44 (sCD44), with survival in nivolumab-treated metastatic non-small cell lung cancer (NSCLC) patients. Plasma biomarkers were assayed at baseline and after two cycles of nivolumab. A cut-off of positivity for sPD-1, sPD-L1 and sCD40L expressions was defined as a plasma level above the lower limit of quantification. Baseline sPD-1 and sPD-L1 levels were subsequently analyzed in a control group of *EGFR*-mutated (*Epidermal Growth Factor Receptor*) NSCLC patients. Association between survival and biomarkers was investigated using Cox proportional hazard regression model. Eighty-seven patients were included (51 nivolumab-treated patients, 36 in EGFR-mutated group). In nivolumab group, baseline sPD-1, sPD-L1 and sCD40L were positive for 15(29.4%), 27(52.9%) and 18(50%) patients, respectively. We defined a composite criteria (sCombo) corresponding to sPD-1 and/or sPD-L1 positivity for each patient. In nivolumab group, baseline sCombo positivity was associated with shorter median progression-free survival (PFS) (78 days 95%CI (55–109) vs. 658 days (222-not reached); HR: 4.12 (1.95–8.71), *p* = 0.0002) and OS (HR: 3.99(1.63–9.80), *p* = 0.003). In multivariate analysis, baseline sCombo independently correlated with PFS (HR: 2.66 (1.17–6.08), *p* = 0.02) but not OS. In EGFR-mutated group, all patients were baseline sCombo positive; therefore this factor was not associated with survival. After two cycles of nivolumab, an increased or stable sPD-1 level independently correlated with longer PFS (HR: 0.49, 95%CI (0.30–0.80), *p* = 0.004) and OS (HR: 0.39, 95%CI (0.21–0.71), *p* = 0.002). VEGFA, sCD40L and sCD44 did not correlate with survival. We propose a composite biomarker using sPD-1and sPDL-1 to predict nivolumab efficacy in NSCLC patients. A larger validation study is warranted.

## 1. Introduction

The advent of immune checkpoint inhibitors has been a turning point in the treatment of metastatic non-small cell lung cancers (NSCLC). Antitumor immunity is indeed considered to be particularly involved in these cancers, due to substantial tobacco-induced DNA damages [1,2]. Programmed cell death 1 (PD-1) pathway inhibitors have drastically changed treatment algorithms for NSCLC patients [3]. Anti-PD-1 monoclonal antibodies bind to PD-1 receptor and prevent interactions with its ligands PD-L1 and PD-L2, thereby releasing PD-1 pathway-mediated inhibition of the immune tumor response [4]. Nivolumab and pembrolizumab are both anti-PD1 therapies approved for the treatment of advanced NSCLC [5,6,7,8].

Anti-PD1 therapies have presented encouraging results during clinical development; however their efficacy remains variable and poorly predictable in daily clinical practice. Even if a small proportion of patients can experiment long responses, the majority of patients does not benefit from these therapies. Thus, the identification of reliable predictive factors is a priority. In regard with pharmacodynamics of anti-PD1 therapies, immunity-related biomarkers such as soluble PD-1 (sPD-1), soluble PD-L1 (sPD-L1), Vascular Endothelial Growth Factor A (VEGFA), soluble CD40L (sCD40L), soluble CD44 (sCD44) could be candidate to predict prognosis and efficacy of anti-PD1 therapies.

The role of sPD-1 is yet to be understood: this variant protein, derived from an alternative splicing of PD-1 messenger RNA (mRNA) is usually undetectable in healthy subjects [9]. High plasma levels of sPD-1 have been positively associated with inflammation levels among pancreatic cancer patients, and with the risk of hepatitis B virus-induced hepatocarcinoma [10,11]. To this date, a single study has analyzed the prognostic role of sPD-1 in NSCLC [12]: In a cohort of 38 *EGFR*-mutated NSCLC patients, an increase in sPD-1 plasma level during erlotinib therapy was associated with a better prognosis. Interestingly, patients with mutation-driven cancer, such as *EGFR*-mutated lung cancer, derive nearly no benefit from immune checkpoint inhibition [13]. Thus, the evaluation of biomarkers in this subset of patient as a comparator is of particular relevance and may be helpful understand the relationship between the biomarkers and patients’ sensitivity to immune checkpoint inhibitors. The expression of sPD-L1 seems to be associated with a poor prognosis in solid cancers [14,15]. This protein can result not only from an alternative splicing of PD-L1 mRNA, but also from a proteolytic cleavage of membrane-bound PD-L1 [16,17]. This marker has been shown to be higher among NSCLC patients in comparison to healthy subjects [18]. Costantini et al. investigated the prognostic role of sPD-L1 in the context of immunotherapy and found that high sPD-L1 expression correlated with shorter overall survival in a cohort of 43 NSCLC patients treated with nivolumab [19]. Furthermore, an increase in sPD-L1 levels after two months of treatment was associated with lower overall response rates. Proangiogenic cytokine VEGFA is known to have an immunosuppressive function in vitro, and induces an increase of PD-1 expression on CD8+ T lymphocytes [20,21]. Studies have suggested that high VEGFA plasma levelsmay be associated with a poor prognosis regardless anticancer agents [22]. In vitro, sCD40L acts as an immunosuppressive factor and induces VEGFA production by endothelial cells [23,24]. However, its prognostic role is yet to be understood [25].Soluble CD44 level in plasma has been positively correlated with tumor burden in gastric cancer patients, and tends to decrease after carcinologic surgery [26]. Salivary and plasmatic detection of sCD44 have been evaluated as potential diagnostic biomarkers for head and neck cancer, but the conclusions are inconsistent [27,28,29]. Finally, preclinical studies suggested that sCD44 may be secreted by tumor cells in triple negative breast cancer and could play a critical role in tumor growth [30]. As the indications for immunotherapies, particularly anti-PD1, are increasing, the need for a reliable biomarker predictive of response has become critical. Tumor PD-L1 expression, used in daily practice, shows limitations in its predictive validity, particularly in consideration of the significant heterogeneity of its expression [31]. Besides, the measurement of tumor mutational burden (TMB) has been recently proposed as a biomarker for immunotherapies in clinical practice. However, TMB also presents limits such as the lack of harmonization in panel-based TMB quantification and of robust predictive cutoffs [32].

The aim of this observational, multicentric study conducted in metastatic NSCLC patients was to assess the prognostic significance of sPD-1, sPD-L1, VEGFA, sCD40L and sCD44 plasma levels at baseline and after two cycles of anti-PD1 therapy nivolumab. Further, we aimed to investigate the predictive significance of sPD-1 and sPD-L1 in comparison with a control cohort of *EGFR*-mutated NSCLC patients, since expected sensitivity to immunotherapy is minimal in this subset of NSCLC.

## 2. Results

### 2.1. Cohorts

One-hundred and twenty-one patients were eligible to this study (Figure 1). Overall, 87 patients were included, with 51 patients in nivolumab group and 36 patients in control *EGFR*-mutated group. *EGFR*-mutated patients were younger (median age 45 (35–51) vs. 66 (60–69) years old, *p* < 0.001). More than 95% of patients had an ECOG performance status of 2 or less in both groups. In nivolumab group, 96% of patients were active or former smokers whereas 69.4% of patients in *EGFR*-mutated group were nonsmokers (Table 1). In nivolumab group, median follow-up was 804 days (553–1112) and median overall survival (OS) and progression-free survival (PFS) were 351 days (304–430) and 132 days (74–350), respectively. Nivolumab therapy was suspended at the discretion of physician in 11 patients for satisfying response (complete or sustained partial response). Among them, only three (27%) had progressed at data cut-off in June 2018 (10.5, 12 and 17 months after nivolumab discontinuation, respectively). At data cut-off, 9 of 51 patients (17.6%) were still treated with nivolumab. In *EGFR*-mutated group, median follow up was 280 days (159–575). Twenty-six patients (72.2%) had progressed at data cut-off with a median PFS of 221 days (110–345).

### 2.2. Biomarker Assessment in the Nivolumab Group

A total of 544 plasma biomarker levels were assayed (Figure 1). In the nivolumab group, among all 51 patients, 27 (52.9%), 30 (58.8%), and 21 (58.3%) patients had baseline quantifiable levels of sPD-1, sPD-L1 and sCD40L, respectively. Baseline VEGFA and sCD44 levels were unquantifiable for only 2 and 4 patients, respectively. There was no significant variation in plasma biomarker levels between baseline and day 28 (Appendix A). Additionally, no correlation was found between biomarkers’ plasma levels and factors such as C-reactive protein (CRP), neutrophil to lymphocyte ratio (NLR), tumor PD-L1 expression, and nivolumab levels regardless sampling occasion (Appendix A).

A cut-off of positivity for sPD-1, sPD-L1 and sCD40L expressions was defined as a plasma level above the lower limit of quantification (0.156 ng/mL). Consequently, baseline sPD-1, sPD-L1 and sCD40L were positive for 15 (29.4%), 27 (52.9%) and 18 patients (50%) in nivolumab group, respectively. As VEGFA and sCD44 were quantifiable for a majority of patients (94%), their plasma levels were considered as continuous variable for the statistical analyses.

### 2.3. Baseline Biomarkers Level and Survival in the Nivolumab Group

In univariate analysis, both baseline expressions of sPD-1 (hazard ratio (HR): 2.59, (95%CI 1.29–5.21), *p* = 0.007) and sPD-L1 (HR: 2.74, (1.38–5.46), *p* = 0.004) were associated with a shorter PFS (Table 2). We defined a composite criteria (sCombo) corresponding to the positivity of sPD-1 and/or sPD-L1 for each patient. In univariate analysis, baseline sCombo positivity correlated with a significant reduction in PFS (78 days, (55–109) vs. 658 days, (222-not reached); HR 4.12, (95%CI 1.95–8.71), *p* = 0.0002) (Figure 2A, Table 2). In multivariate analysis, baseline sCombo positivity was independently associated with a shorter PFS (HR: 2.66, (1.17–6.08), *p* = 0.02) whereas tumor PD-L1 expression was not (HR: 0.99 (0.98–1.00) *p* = 0.051).

Overall survival was decreased in patients tested positive for sCombo (median OS: 367 days (167–501) vs. not reached (402–not reached); HR: 3.99, (95%CI 1.63–9.80), *p* = 0.003) (Figure 2B, Table 2). However, in multivariate analysis, only tumor PD-L1 expression rate was independently associated with OS (multivariate HR: 0.98, (95%CI 0.97–0.99), *p* = 0.043; Table 2).

Overall, sCombo-positive patients were at higher risk of treatment failure (67.7% vs. 30.0%, *p* = 0.011). Interestingly, a single patient was sCombo-positive among the 11 long-term responding patients (Appendix A presents best objective response (BOR) according to sCombo status).

Baseline VEGFA, sCD40L and sCD44 plasma levels did not correlate with PFS, OS, or BOR. Outcomes of patients with a positive expression of sCD40L did not differ from those without detectable sCD40L expression (Table 2).

### 2.4. Baseline Biomarkers among EGFR-Mutated NSCLC Patients

In the control group of *EGFR*-mutated NSCLC patients, baseline sPD-1 expression was positive for 15 patients (41.7%). Furthermore, all patients had detectable baseline sPD-L1 levels (median IQR): 1.50 ng/mL (0.70–2.22)) and therefore were sCombo positive. Mean sPD-L1 level was about 5 fold higher in control group compared to nivolumab group (1.67+/-0.91 g/mL vs. 0.25 +/-0.22 ng/mL, *p* < 0.001). However, levels of sPD-L1 did not seem to correlate with patients’ PFS (HR: 0.87 (0.60–1.27), *p* = 0.47). Baseline sPD-1 positivity did not correlate with patients’ PFS (HR 2.05 (0.93–4.52), *p* = 0.75).

### 2.5. Biomarkers Kinetics and Survival in Nivolumab Group

At day 28, none of the biomarkers were independently associated with PFS or OS. Interestingly, an increased (>30%) or stable sPD-1 level after two cycles of nivolumab was independently associated with a longer PFS and OS in comparison with patients harboring decreased levels of sPD-1 (median PFS: 121 days(78–320) vs. 50 days (36–not reached), multivariate HR: 0.49, 95%CI (0.30–0.80), *p* = 0.004; median OS 450 days (386-not reached) vs. 153 days (68-not reached), multivariate HR: 0.39, (95%CI 0.21–0.71), *p* = 0.002) (Figure 3, Table 3). Soluble PD-L1 reduction after two cycles was associated with a poor prognosis in univariate analysis, but this result was not confirmed in multivariate analysis.

The evolution of VEGFA, sCD40L and sCD44 concentrations between baseline and day 28 did not predict patients’ outcomes (Table 3).

## 3. Discussion

The search for reliable biomarkers to predict the efficacy of immune checkpoints inhibitors is a major pathway of treatment optimization. Indeed, identifying non responders would prevent patients from the loss of opportunities for more efficient therapies, but also limit the financial burden of unnecessary treatments [33]. To this day, research has mainly focused on tumor PD-L1 expression for this purpose. However, its predictive value remains debated, since it is subject to significant temporal and spatial variability [34,35,36]. Recently, several studies have highlighted the value of tumor mutational burden (TMB) for the prediction of checkpoint inhibitors’ efficacy in NSCLC patients [37,38]. Moreover, Gandara et al. developed a noninvasive blood-based assay to measure TMB on circulating tumor DNA, thus limiting biopsy-related sampling bias. They confirmed a significant improvement of progression-free survival (PFS) rates in metastatic NSCLC patients harboring high TMB levels treated with anti-PDL1 therapy atezolizumab [39]. Although promising, TMB assessment requires substantial tumor sample, access to a genomic profiling platform and financial support to assume the cost of whole exome sequencing in a wide scale. Finally, kinomic and genomic approaches were also proposed to predict efficacy of anti-PD1 therapy in NSCLC, however results are too preliminary to be used in daily clinical practice [40]. Thus, the need remains for a more practical and cost-effective biomarker. The present study suggests that baseline expression of at least one biomarker among sPD-1 and sPD-L1 (referred to as sCombo positivity) in NSCLC patients treated with nivolumab may be predictive of treatment failure.

We investigated the prognostic value of five plasmatic biomarkers, sPD-1, sPD-L1, VEGFA, sCD40L and sCD44, before and after introduction of nivolumab in NSCLC patients. Baseline sPD-1 and sPD-L1 were further investigated in a population of *EGFR*-mutated NSCLC. As far as we know, this is the first study questioning the impact of these biomarkers both in immune-sensitive and in immune-resistant populations. First, we identified a composite biomarker sCombo as an independent prognostic factor associated with shorter PFS among patients treated with nivolumab. We subsequently investigated this biomarker in a cohort of 36 *EGFR*-mutated patients. Previous studies have reported that *EGFR*-mutated NSCLC patients are poor responders to nivolumab [13,41]. All 36 patients exhibited baseline sCombo positivity due to a constant expression of sPD-L1, thus, these results suggest that baseline sCombo positivity may not only be prognosic, but most importantly could be predictive of failure of nivolumab therapy in NSCLC patients. Second, we showed that a decrease in sPD-1 concentration after two cycles of nivolumab was associated with a reduction of both PFS and OS. Assessing sPD-1 kinetics between baseline and day 28 reflects the impact of nivolumab on the biomarker’s production or destruction and may be helpful to identify non responders before the first radiological evaluation, which usually occurs after 4 to 6 cycles. Finally, we demonstrated that baseline sCombo positivity and sPD-1 kinetics are more reliable factors in this cohort than baseline tumor PD-L1 expression. These results highlight once again the weakness of tumor PD-L1 expression as a predictive biomarker. Indeed, the optimal cut-off of 10% in our study differs from those of previous publications, which points out the challenge of defining an universal limit to efficiently identify patients likely to respond to immune checkpoint inhibition [8,42].

Data about the role of sPD-1 and sPD-L1in anti-tumor immune response in NSCLC remain currently sparse [43]. In vivo studies have suggested that sPD-1 could interact with membrane-bound PD-L1 and PD-L2 (mPD-L1, mPD-L2) and might enhance immunity by limiting the interaction of mPD-1 with these ligands [44,45]. In these studies, the introduction of vectors expressing only the soluble form of PD-1 protein in tumor-bearing mice resulted in an increase of T lymphocyte proliferation, an up-regulation of prionflammatory cytokines, co-stimulatory molecules and antitumor response. The proimmunologic role of sPD-1 was also suggested in autoimmune diseases such as rheumatoid arthritis. In these cases, levels of sPD-1 in plasma and synovial fluid positively correlated with titers of rheumatoid factor and concentrations of TNF alpha, and increased levels of sPD-1 in plasma and synovial fluid were associated with higher inflammation and disease activity [46,47]. Based on these results, the pejorative role of baseline sPD-1 positivity in our study suggests that sPD-1 may prevent interaction between membrane-bound PD-1 (mPD-1) and mPD-L1/ mPD-L2. Thus, sPD-1 would limit mPD-1 mediated co-inhibitory signal on T CD8 lymphocytes and therefore would be associated with improved anti-tumor immune response. In this context, the benefit of nivolumab therapy may be lower for patients expressing sPD-1 before treatment initiation (Figure 4). Furthermore, the correlation between sPD-1 kinetics and patients’ prognosis under nivolumab therapy could be a consequence of interindividual variability in sPD-1 sensitivity to nivolumab. Indeed, studies have highlighted the impact of mPD-1 N-glycosylation and terminal N-loop on nivolumab’s affinity [48,49]. Among the four PD-1 splice variants identified, only one (PD-1Δex3) seems to have the complete structure of PD-1 extracellular domain (including PD-L1/PD-L2 binding domain) corresponding to sPD-1 [9]. Hence, it is likely that post-traductional modifications of sPD-1 N-terminal domain could alter its affinity fornivolumab, which would reflect through sPD-1 evolution at day 28. Consequently, a decrease of sPD-1 levels may be a sign of sPD-1 targeting by nivolumab, and one could expect a pejorative outcome in this situation: first due to the loss of sPD-1-induced immune activity, but also because of an inevitable competition between sPD-1 and mPD-1 for nivolumab binding. Conversely, if sPD-1 exhibited low affinity for nivolumab, the drug would mostly target mPD-1, resulting in a better antitumor effect. As aforementioned, Sorensen et al. demonstrated that an increase of sPD-1 concentrations between baseline and progression among 38 *EGFR*-mutated NSCLC patients was associated with a better prognosis. This result falls in line with our hypothesis, since these patients did not receive anti-PD1 therapy, therefore may have additively benefited from sPD-1 proimmunologic effect [12].

In vitro experimentations have shown that sPD-L1 is mainly released by tumor cells and mature dendritic cells and can induce apoptosis of CD4+ and CD8+ T lymphocytes [50]. Numerous clinical studies have demonstrated that sPD-L1 expression negatively correlate with prognosis [14,19,51,52,53,54]. Recently, a functional ELISA test has been developed to specifically detect sPD-L1 with PD-1 binding capacity. Authors have shown that glycosylation of sPD-L1 could substantially modify its binding capacities, and that sPD-L1 with binding capacity was detected in plasma of NSCLC patients [55]. These results suggest thatsPD-L1 may bind to T lymphocytes’mPD-1 and mimic mPD-L1 ability to induce lymphocyte exhaustion, which would contribute to tumor immune evasion. Second, a competition between sPD-L1 and nivolumab for mPD-1binding may reducethe pharmacodynamic activity of nivolumab and therefore its efficacy. In our control cohort of 36 NSCLC *EGFR*-mutated patients, all patients exhibited a constant baseline expression of sPD-L1. Given that *EGFR*-mutated lung cancers are known to be resistant to anti-PD1 therapies [41,56], these findings support a deleterious effect of sPD-L1 expression on nivolumab efficacy in NSCLC patients. Interestingly, in a previous study a similar trend has been observed among NSCLC patients treated with another anti-PD1, pembrolizumab combined with low doses of chemotherapy. Patients who manifested progressive disease tended to have higher levels of sPD-L1 than responders, but differences were not statistically significant and patients were not treated with pembrolizumab in monotherapy [57]. Moreover, sPD-L1 was also associated with a poor prognosis among patients treated with anti-CTLA4 for a melanoma, suggesting that the implications of this soluble biomarker extend beyond its interactions with mPD-1/PD-L1 [54]. Taken together these results suggest that the prognostic value of sPD-L1 should be further investigated regardless the mechanism of action of immunotherapy.

Despites appealing results, this study has some limitations. First, conclusions should be drawn with caution in this small cohort. Only 40 patients could be included in the multivariate analyses in nivolumab group, mainly because of missing data regarding tumor PD-L1 expression. However, the magnitude of sCombo and sPD-1 kinetics correlation to patients’ outcomes encourages further investigations regarding their predictive significance in larger populations. Second, we used ELISAs from the same manufacturers for all patients in our study, so the possibility of assay-dependent variability was not addressed. The lack of comparison between tests suggests that the numerical threshold defined in our study should be validated if another test is used. Besides, the test used did not provide any functional indication about the binding capacity of the biomarkers detected. Third, we investigated *EGFR*-mutated NSCLC patients as a control cohort for this study. Thus, baseline patients’ characteristics differed between cases and controls. It was considered to be the best choice given that most patients with *EGFR* wild-type NSCLC would receive anti-PD1 therapy during the course of their diseases. Additionally, *EGFR* mutation is known to be associated with resistance to immunotherapy, therefore evaluating biomarkers in these patients is of particular interest [41]. Interestingly, studies have shown that *EGFR*-mutated NSCLC tended to express PD-L1 on tumor cells at higher frequency, yet they benefit less from anti-PD1 therapies than wild-type tumors [13,58,59]. Although tumor PD-L1 expression could not be assessed in our control *EGFR*-mutated group due to a lack of available tumor samples, these information suggest that sPD-1 and sPD-L1 may play a part in resistance to anti-PD1 therapies in this specific population. Furthermore, sPD-1 kinetics could not be assessed in the control *EGFR*-mutated group. Therefore, its predictive significance should be further investigated. Finally, sCombo does not seem to correlate with overall survival in multivariate analysis among patients treated with nivolumab. This may result from a lack of power in our study. In this limited cohort, heterogeneous choices of treatments subsequent to nivolumab failure may have impacted the statistical interpretation of overall survival data.

## 4. Patients and Methods

### 4.1. Patients

This observational multicentric study included patients from three French medical centers (Cochin Hospital, Georges Pompidou European Hospital, Ambroise Paré Hospital) treated between July 2015 and June 2018. First, consecutive patients with metastatic lung cancer undergoing nivolumab, which were all part of the CERTIM prospective cohort (Immuno-modulatory Therapies Multidisciplinary Study group, Cochin Hospital, Paris, France) were screened. Inclusion criteria were the administration of at least two cycles of nivolumab, availability of plasma samples at baseline (less than two weeks before the first injection of nivolumab) and at day 28 just before the third infusion. Exclusion criteria were small-cell cancer or a mixed tumor with a neuroendocrine small-cell component.

Henceforth, a control group of *EGFR*-mutated NSCLC patients was retrospectively established to estimate the predictive value of baseline sPD-1 and sPD-L1 levels regarding survival. Patients of the control group were enrolled from three medical centers (Cochin hospital, Georges Pompidou European Hospital, Ambroise Paré Hospital). Inclusion criteria were: metastatic NCSLC with a targetable *EGFR* somatic mutation receiving a specific tyrosine kinase inhibitor; no prior or current nivolumab therapy. Patients were excluded if they were unfit to receive anti-*EGFR* therapy after baseline assessment of biomarkers.

All patients had signed a written informed consent before the beginning of this study. This project was in compliance with the Declaration of Helsinki and approved by the local medical ethical board.

### 4.2. Nivolumab Treatment

Patients in nivolumab group were treated with the recommended dose of nivolumab (3 mg/kg every two weeks) administered intravenously over 60 min. All adverse events were graded using the National Cancer Institute Common Toxicity Criteria, version 4.0. Nivolumab treatment was continued until disease progression (either based on clinical or modified Radiological Response Evaluation Criteria in Solid Tumors version 1.1), or unacceptable toxicity. However, treatment temporary discontinuation was authorized in the cases of significant response; the timing of such interruption was left to the discretion of physicians.

### 4.3. Blood Collection

Venous whole blood samples were collected in lithium heparin tubes at baseline (within a month before nivolumab or *EGFR* inhibitor initiation) and just before the third infusion for patients undergoing nivolumab therapy. Plasma was separated by centrifugation (4000 rpm, 10 min, 4 °C) within 2 h after blood collection, and then stored at −20 °C until the analysis. Nivolumab concentrations were assessed on all samples using an ELISA method we previously validated [60]. For this study, plasma samples were defrosted and centrifuged (1000 rpm, 10 min) before dosing.

### 4.4. Plasma Levels of Biomarkers

Plasma levels of soluble PD-1, sPD-L1 and sCD40L were assayed using ELISA kits SEA751Hu, SEA788Hu and SEA119Hu (Cloud-Clone Corp., TX, USA). Their assay range was 0.156 – 10 ng/mL with detection limits of 0.063 ng/mL, 0.057 ng/mL and 0.064 ng/mL, respectively. Soluble CD44 plasma levels were assayed using ELISA kit SEA670Hu (Cloud-Clone Corp). The assay range was 1.56–100 ng/mL. ELISA kit DVE00 (R*&D* Systems, MI, USA) was used to determine plasma VEGFA levels. Its assay range was 31.5–2000 pg/mL with a detection limit of 9 pg/mL. For all these tests, intra-assay and inter-assay precision were inferior to 10% and 12%, respectively. All plasma levels of biomarkers were assayed in duplicate according to manufacturer’s instructions and a random patient’s sample was tested on two different plates to ensure the reproducibility of the method.

### 4.5. Clinical Endpoints

The main objective was to explore the relationship between plasma levels of biomarkers and survival. The primary end-point was PFS, defined as the time from treatment initiation (nivolumab or anti-*EGFR*) to documented clinical or radiological progression event or death from any cause. Radiological evidence of progression was defined according to modified Response Evaluation Criteria in Solid Tumors1.1 (RECIST 1.1) [61].

Secondary end-points were OS, defined as the time from treatment initiation to death from any cause and best objective response (BOR) according to RECIST 1.1.

### 4.6. Statistics

Descriptive statistics used median (interquartile range) for quantitative variables and percentages for qualitative ones. PFS and OS are presented as median (95% confidence interval (95% CI)). The correlation between variables including plasma biomarker levels was estimated using a Spearman test. In nivolumab group, the kinetics of all biomarkers was characterized by their evolution between baseline and day 28. A paired Wilcoxon test was used to compare baseline plasma levels of each biomarker to those at day 28 (Delta D28–D0). For each biomarker, the minimal level of variation considered as significant was 30%, provided that at least one of the concentrations was above the lower limit of quantification. In cases of a variation below 30% or if the concentrations at baseline and day 28 were both unquantifiable, the levels of biomarkers were considered to be steady between baseline and day 28. This 30% margin of error was set to take into account a 10% risk of analytical variability and the expected physiological variability of the biomarkers’ concentrations. Survival data were analyzed using Kaplan-Meier curves and a Cox proportional hazard regression model adjusted for established clinical and biological risk factors. Multivariate analyses were performed for biomarkers and factors with a *p*-value lower than 0.1 and data considered as clinically meaningful, such as Eastern Cooperative Oncology Group Performance Status (ECOG PS). *p*-value was set to 0.05. Statistical analyses were performed using R program (The R Foundation for Statistical Computing, Vienna, Austria).

## 5. Conclusions

In conclusion, this pilot study paves the way towards a potential predictive role of baseline sCombo (sPD-1 and/or sPD-L1 expression) for nivolumab efficacy. Larger-scale confirmation studies will be needed to assess the place of these noninvasive and easily assessed biomarkers in clinical daily practice.

This work was presented at American Association for Cancer Research (AACR) annual meeting 2019 at Atlanta (Abstract #4107).

## Figures and Tables

**Figure 1 cancers-12-00473-f001:**
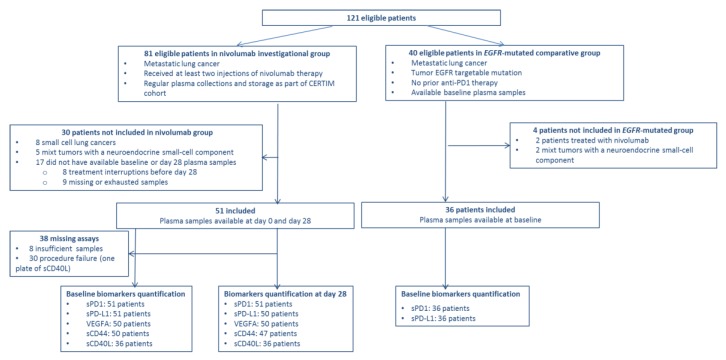
Flow chart.

**Figure 2 cancers-12-00473-f002:**
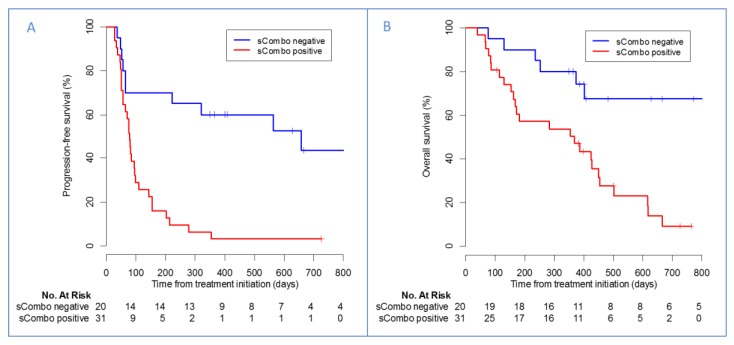
Kaplan–Meier estimates of patients’ (**A**) progression-free survival and (**B**) overall survival according to sCombo status.

**Figure 3 cancers-12-00473-f003:**
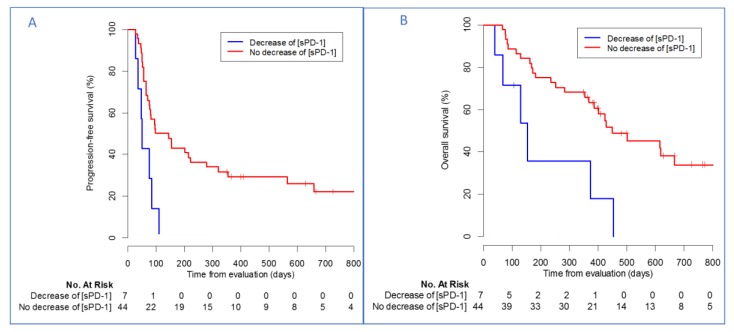
Kaplan–Meier estimates of patients’ (**A**) progression-free survival and (**B**) overall survival according to sPD-1 evolution between baseline and day 28. Tick marks indicate censoring of data.

**Figure 4 cancers-12-00473-f004:**
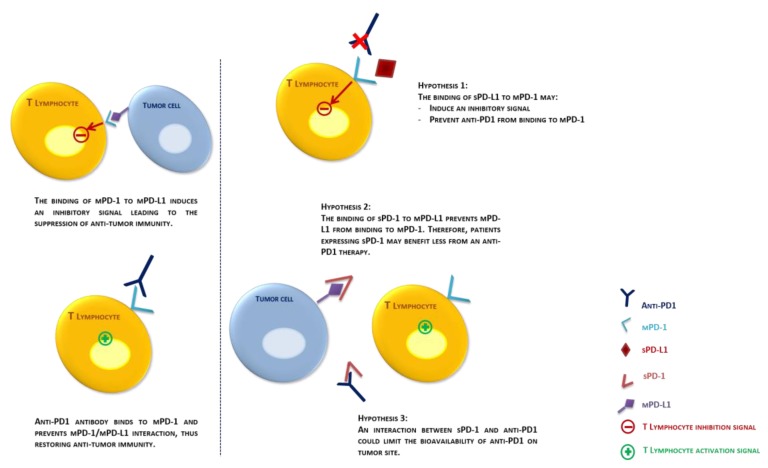
Three theoretical interaction between sPD-1, sPD-L1 and the tumour and its environment, leading to resistance to anti-PD1 therapies. sPD-1/ sPD-L1: soluble PD-1/PD-L1; mPD-1/mPD-L1: membrane-bound PD-1/ PD-L1.

**Table 1 cancers-12-00473-t001:** Patients’ characteristics.

Characteristics	Nivolumab (n = 51)	EGFR-Mutated (n = 36)
**Sex, n (%)**		
**Males**	29 (56.9)	7 (19.4)
**Females**	22 (43.1)	29 (80.6)
**Median age, years (IQR)**	66 (60–69)	45 (36–54)
**Smoking habit, n (%)**		
**Never**	2 (4)	25 (69.4)
**Former**	37 (72.5)	10 (27.8)
**Active**	12 (23.5)	1 (2.8)
**ECOG, n (%)**		
**0**	1 (2)	9 (25)
**1**	29 (56.8)	19 (52.8)
**2**	19 (37.2)	8 (22.2)
**3**	2 (4)	0 (0)
**Type of cancer, n (%)**		
**Adenocarcinoma**	40 (78.4)	34 (94.4)
**Squamous cell carcinoma**	11 (21.6)	0 (0)
**Large cell**	0 (0)	2 (5.6)
**Metastases, n (%)**		
**Synchronous**	35 (68.6)	30 (83.3)
**Metachronous**	16 (31.4)	6 (16.7)
**Number of metastatic sites, n (%)**		
**1**	21 (41.2)	18 (50)
**2**	15 (29.4)	4 (11.1)
**3**	11 (21.6)	9 (25)
**> 3**	4 (7.8)	5 (13.9)
**Mutations, n (%)**		
***EGFR***	2 (4)	36 (100)
***ALK***	2 (4)	0 (0)
***KRAS***	12 (23.5)	0 (0)
**Others**	4 (7.8)	0 (0)
**Tumor PD-L1 expression, n (%)**		
**0**	14 (27.5)	-
**1–4%**	3 (5.9)	-
**5–9%**	6 (11.7)	-
**10–24%**	5 (9.8)	-
**25–49%**	3 (5.9)	-
**≥ 50%**	(17.6)	-
**Unknown**	11 (21.6)	-
**Prior antiangiogenic therapy, n (%)**		
**Yes**	15 (29.4)	0 (0)
**No**	36 (70.6)	36 (100)
**Number of prior lines before nivolumab, n (%)**		
**1**	35 (68.6)	-
**2**	7 (13.7)	-
**3**	6 (11.8)	-
**> 3**	3 (5.9)	-

ECOG: Eastern Cooperative Oncology Group.

**Table 2 cancers-12-00473-t002:** Univariate and multivariate Cox proportional hazard analyses of progression-free survival (PFS) and overall survival (OS) according to baseline plasma biomarker expression among patients treated with nivolumab.

VARIABLES	PFS HR (95%CI) Univariate	*p*	PFS HR (95%CI) Multivariate	*p*	OS HR (95%IC) Univariate	*p*	OS HR (95%IC) Multivariate	*p*
**Patients’ characteristics (n = 51)**								
Age	1.00 (0.96–1.04)	0.899			0.99(0.94–1.04)	0.695		
ECOG 2 or 3 vs. 0 or 1	1.09 (0.59–2.00)	0.783	1.23 (0.59–2.57)	0.583	1.13 (0.56–2.27)	0.733	1.03 (0.45–2.36)	0.944
Smoking: former vs. never	0.65 (0.15–2.73)	0.552			0.66 (0.09–4.93)	0.683		
Smoking: active vs. never	0.59 (0.13–2.77)	0.504			0.53 (0.06–4.51)	0.564		
**Biology (n = 51)**								
CRP	1.01 (1.00–1.02)	**0.029**	1.00 [0.99–1.02]	0.578	1.01 (0.99–1.02)	0.309	0.99 (0.98–1.01)	0.709
NLR	1.06 (0.98–1.15)	0.146			1.12 (1.02–1.24)	**0.020**		
Lymphocytes	1.00 (0.99–1.00)	0.132			0.99 (0.99–1.00)	**0.012**		
**Pathology (n = 40)**								
PD–L1 TC	0.99 (0.98–1.00)	0.067	0.99 [0.98–1.00]	0.051	0.99 (0.97–0.99)	**0.060**	0.99 (0.97–1.00)	0.058
PD–L1 TC: 1% cut–off	0.62 (0.31–1.22)	0.166			0.62 (0.29–1.36)	0.236		
PD-L1 TC: 10% cut-off	0.47 (0.23–0.97)	**0.040**			0.35 (0.14–0.85)	**0.021**		
PD-L1 TC: 50% cut-off	0.51 (0.21–1.24)	0.136			0.47 (0.16–1.38)	0.170		
**Plasma biomarkers**								
**sPD-1**	3.03 (1.01–9.12)	**0.049**			2.33 (0.76–7.18)	0.141		
sPD-1 positive (n = 15) vs. negative (n = 36)	2.59 (1.29–5.21)	**0.007**			2.28 (1.11–4.68)	**0.025**		
**sPD-L1**	1.89 (0.72–4.95)	0.194			2.15 (0.77–6.01)	0.145		
sPD-L1 positive (n = 27) vs. negative (n = 24)	2.68 (1.36–5.28)	**0.004**			2.68 (1.23–5.84)	**0.013**		
**sPD-1 and/or sPD-L1**								
1 positive biomarker (n = 20) vs. 0 (n = 20)	4.13 (1.89–9.02)	**0.0004**			4.00 (1.54–10.40)	**0.004**		
2 positive biomarkers (n = 11) vs. 0 (n = 20)	4.11 (1.64–10.3)	**0.003**			3.99 (1.44–11.00)	**0.008**		
**sCombo positive (n = 31) vs. negative (n = 20)**	4.12 (1.95–8.71)	**0.0002**	2.66 [1.17–6.08]	**0.020**	3.99 (1.63–9.80)	**0.003**	2.17 (0.86–5.45)	0.101
**VEGFA**	1.00 (1.00–1.00)	0.17			1.00 (0.99–1.00)	0.525		
**sCD44**	1.02 (0.97–1.07)	0.399			1.01 (0.96–1.07)	0.650		
**sCD40L**	2.04 (0.56–7.97)	0.278			1.00 (0.99–1.00)	0.515		
sCD40L positive (n = 13) vs. negative (n = 23)	1.43 (0.67–3.03)	0.355			1.59 (0.35–7.14)	0.546		

95% CI: 95% confidence interval; CRP: C-reactive protein; ECOG: Eastern Cooperative Oncology Group Performance Status;HR: Hazard Ratio; PD-L1 TC: Tumor PD-L1 expression; NLR, Neutrophil to Lymphocyte Ratio; sCombo: sPD-1 and/or sPD-L1 expression; VEGF: Vascular Endothelial Growth Factor A.

**Table 3 cancers-12-00473-t003:** Univariate and multivariate Cox proportional hazard analyses of progression-free survival (PFS) and overall survival (OS) according to plasma biomarker evolution between baseline and day 28 among patients treated with nivolumab.

VARIABLES	PFS HR (95%CI) Univariate	*p*	PFS HR (95%CI) Multivariate	*p*	OS HR (95%CI) Univariate	*p*	OS HR (95%CI) Multivariate	*p*
**ECOG 2 or 3 vs. 0 or 1 (n = 51)**	1.09 (0.59–2.00)	0.783	1.08 (0.53–2.23)	0.827	1.13 (0.56–2.27)	0.733	0.78 (0.33–1.83)	0.566
**CRP**	1.01 (1.00–1.02)	**0.029**	1.01 (0.99–1.02)	0.230	1.01 (0.99–1.02)	0.309	1.00 (0.99–1.02)	0.878
**PD-L1 TC (n = 40)**	0.99 (0.98–1.00)	0.067	0.99 (0.98–1.00)	0.058	0.99 (0.97–0.99)	0.060	0.99 (0.97–1.00)	0.075
**Delta sPD-1**								
Positive or null (n = 44) vs. negative (n = 7)	0.29 (0.12–0.68)	**0.005**	0.49 (0.30–0.80]	**0.004**	0.28 (0.11–0.70)	**0.007**	0.39 (0.21–0.71)	**0.002**
Positive (n = 9) vs. null or negative (n = 42)	0.93 (0.429–2.02)	0.855			0.71 (0.29–1.73)	0.447		
**Delta sPD-L1**								
Positive or null (n = 44) vs. negative (n = 7)	0.69 (0.34–1.38)	0.291			0.41 (0.20–0.87)	**0.020**		
Positive (n = 7) vs. null or negative (n = 43)	1.05 (0.44–2.51)	0.908			0.59 (0.19–1.94)	0.383		
**Delta VEGFA**								
Positive or null (n = 38) vs. negative (n = 12	1.15 (0.55–2.42)	0.709			1.16 (0.52–2.58)	0.725		
Positive (n = 15) vs. null or negative (n = 35)	0.71 (0.36–1.43)	0.338			0.91 (0.42–1.97)	0.803		
**Delta sCD44**								
Positive or null (n = 38) vs. negative (n = 8)	0.90 (0.39–2.05)	0.796			0.87 (0.30–2.52)	0.797		
Positive (n = 18) vs. null or negative (n = 28)	0.53 (0.26–1.06)	0.074			0.55(0.26–1.19)	0.129		
**Delta sCD40L**								
Positive or null (n = 26) vs. negative (n = 7)	0.94 (0.39–2.25)	0.883			0.71 (0.26–1.97)	0.509		
Positive (n = 10) vs. null or negative (n = 23)	1.07 (0.47–2.43)	0.869			1.14 (0.48–2.68)	0.766		

95% CI: 95% confidence interval; CRP: C-reactive protein; ECOG: Eastern Cooperative Oncology Group; HR: Hazard Ratio; Delta (D28-D0): Differences of concentration at day 28 and baseline; PD-L1 TC: Tumor PD-L1 expression; VEGF: Vascular Endothelial Growth Factor A.

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
