# Peer review of "Predictive Value of Soluble PD-1, PD-L1, VEGFA, CD40 Ligand and CD44 for Nivolumab Therapy in Advanced Non-Small Cell Lung Cancer: A Case-Control Study"

_cancers, 2020, doi:10.3390/cancers12020473_

Round 1

Reviewer 1 Report

Manuscript titled: ‘Predictive value of soluble PD-1, PD-L1, VEGFA, CD40 ligand and CD44 for nivolumab therapy in advanced non-small cell lung cancer: a case-control study’ is very interested work.

Manuscript is about soluble PD-1 (sPD-1), soluble PD-L1 (sPD-L1), VEGFA, sCD40L and sCD44, in nivolumab-treated metastatic NSCLC patients. It is very important topic due to the lack of effectiveness of immunotherapy in patients with high tumor PD-L1 protein level and inversely - anti-PD-/anti-PD-L1 treatment efficacy is observed without PD-L1 expression on tumor cells. It seems that for assessment of immunotherapy efficacy, auxiliary markers are needed for the PD-L1 tumor expression. The Authors suggested a composite biomarker using sPD-1and sPDL-1 to predict nivolumab efficacy in NSCLC patients.

Comments to the article below.

Figures and tables (except Table 1) are blurred (poor resolution?). I am thinking about the EGFR-mutated group. Maybe it would be worth adding clearly that this is a comparative group for the examined cohort of immunotherapy treated patients? In Discussion section: ‘In this studies, the introduction of sPD-1 gene in tumor-bearing mice resulted…’. Isn't it that the PD-L1 gene is one for membrane PD-L1 and for soluble form? Moreover names of genes should be write in italics. Shouldn't the Conclusion section be after the Discussion section? And finally, is it possible to replace the word ‘Plus’ at the beginning of a few sentences trough the manuscript with some other words? We have ‘However’, ‘Moreover’ and a few others that will be adequate and definitely improves impression of the reader.

Reviewer 2 Report

This manuscript presents an interesting study in the search for biomarkers to assess and predict response rate to checkpoint inhibition by nivolumab in NSCLC. Therefore it’s positive findings on sPD1 and sPDL1 are important, as are the negative conclusions for the other markers (VEGFA, CD40L and CD44). It is expecially well recognized, that the authors are cautious in drawing conclusions due to small patient cohorts and the differences in patient characteristics between nivolumab and EGFR-mutated groups. Interestingly the newer references and justification for choosing those patient gourps are mainly found in the discussion. The introduction must be improved substancially and would benefit from some arguments presented only late in the discussion.

Even though this reviewer recognizes the time needed for setting up and performing a clinical study, in this extremely fast developing field of checkpoint inhibition literature form the last year needs to be taken into account. There are only few references dating 2018 (in the discussion) and only one (self-) reference for 2019. Especially papers refering to difficulties and reliabilty of measuring soluble markers used here should be discussed.

The rational for EGFR-mutated NSCLC as control group and immune-resistant (see discussion) should be elaborated more and early in the manuscript for readers who are not experts in NSCLC (even though it seems to be adressed in reference 36, again cited only in the discussion). This is especially important the patients‘ characteristics (given in Table 1) are quite different (in sex, smoking, tumor type and number of metastatic sites) and mutational status for the nivolumab group does not add up to half oft he patients.

Has PD-L1 tumor expression in the EGFR-mutated group not been assesed (why?). This should be commented on, since in all those patients baseline sPD-L1 is stated as positive.

May an age dependent expression of sPD1 play a role for the baseline measurements?

In Supplementary Figure 1 marker levels are displayed for all individual patients. How can there be values below detection limits?

In the discussion the auhors elaborate on a mechanism for sPD1 and sPD-L1. A graphic depiction oft the model could be helpfull.

It is recognized by this reviewer, that this study focused on one checkpoint inhibitor (nivolumab), yet given the clinical importance, a (speculative) outlook to other checkpiont inhibitors (and anti-PDL1 or combination treatments) would be interesting.

Minor points:

Reference 1 does not refer to tumor immunology, while reference 2 is a review from 2013 on tumor immunology, but both papers do not reference tobacco induced DNA damages. Introduction leading to reference 12: Which study has ‚questionned‘ the value of sPD1 (or should it be ‚analyzed‘)? References justifing, CD44 as biomarkers are dating quite far back. In Point 2.5 the statement in the first sentence does seem to contradict the following passages and Table 3. Reference list must be in english (month and some other words are in French)
